# Selected Pharmaceuticals in Different Aquatic Compartments: Part II—Toxicity and Environmental Risk Assessment

**DOI:** 10.3390/molecules25081796

**Published:** 2020-04-14

**Authors:** André Pereira, Liliana Silva, Célia Laranjeiro, Celeste Lino, Angelina Pena

**Affiliations:** LAQV, REQUIMTE, Laboratory of Bromatology and Pharmacognosy, Faculty of Pharmacy, University of Coimbra, Polo III, Azinhaga de St Comba, 3000-548 Coimbra, Portugal; ljgsilva@hotmail.com (L.S.); celialaranjeiro@gmail.com (C.L.); cmlino@ci.uc.pt (C.L.); apena@ci.uc.pt (A.P.)

**Keywords:** environmental contaminants, pharmaceuticals, pharmaceuticals toxicity, environmental risk assessment, aquatic compartments

## Abstract

Potential risks associated with releases of human pharmaceuticals into the environment have become an increasingly important issue in environmental health. This concern has been driven by the widespread detection of pharmaceuticals in all aquatic compartments. Therefore, 22 pharmaceuticals, 6 metabolites and transformation products, belonging to 7 therapeutic groups, were selected to perform a review on their toxicity and environmental risk assessment (ERA) in different aquatic compartments, important issues to tackle the water framework directive (WFD). The toxicity data collected reported, with the exception of anxiolytics, at least one toxicity value for concentrations below 1 µg L^−1^. The results obtained for the ERA revealed risk quotients (RQs) higher than 1 in all the aquatic bodies and for the three trophic levels, algae, invertebrates and fish, posing ecotoxicological pressure in all of these compartments. The therapeutic groups with higher RQs were hormones, antiepileptics, anti-inflammatories and antibiotics. Unsurprisingly, RQs values were highest in wastewaters, however, less contaminated water bodies such as groundwaters still presented maximum values up to 91,150 regarding 17α-ethinylestradiol in fish. Overall, these results present an important input for setting prioritizing measures and sustainable strategies, minimizing their impact in the aquatic environment.

## 1. Introduction

The environmental impact of medicinal products has been recognized worldwide, and as its use cannot be avoided, a sound risk assessment of their presence in the environment is a key issue that must be tackled to meet the European Union (EU) Water Framework Directive (WFD) [1]. The potential of human pharmaceuticals for negative ecotoxicological effects, even at sublethal concentrations, in the aquatic environment has been of concern since the issue was first brought to attention in 1985 [2]. Nonetheless, the ecotoxicological risks associated to the ubiquitous occurrence of pharmaceuticals in aquatic ecosystems are far from being fully known [3].

According to the European Medicines Agency (EMA) legislation, and since 2006, before a pharmaceutical obtains a marketing authorization approval, it must be demonstrated that it poses no risk to the environment through an environmental risk assessment (ERA). ERA compares the predicted environmental concentrations (PECs), with the predicted no effect concentrations (PNECs) of three trophic levels of aquatic organisms [4,5]. When the pharmaceutical is already on the market, instead of using the PEC, which predict the environmental concentration, we can use the measured environmental concentrations (MEC) reflecting the real concentration in the aquatic environment [6]. Therefore, for marketed pharmaceuticals, high-quality monitoring data, along with data on ecotoxicological and toxicological effects are crucial to perform the ERA, which associates the presence of pharmaceuticals with their impact on the aquatic ecosystem and human health, supporting the selection of possible new priority substances to be monitored [7,8,9].

Thus, a systematic review, in order to provide a clear insight on pharmaceuticals toxicology and on ERA, should embrace, not only several parent compounds, but also, metabolites and transformations products belonging to different therapeutic groups such as: the anxiolytics and hypnotics, further referred only as anxiolytics, alprazolam (ALP), lorazepam (LOR) and zolpidem (ZOL); the antibiotics azithromycin (AZI), ciprofloxacin (CIP), clarithromycin (CLA) and erythromycin (ERY); the lipid regulators bezafibrate (BEZ), gemfibrozil (GEM) and simvastatin (SIM); the antiepileptic carbamazepine (CAR); the selective serotonin reuptake inhibitors (SSRIs) citalopram (CIT) and its main metabolite desmethylcitalopram (N-CIT), escitalopram (ESC), fluoxetine (FLU) and its main metabolite norfluoxetine (Nor-FLU), paroxetine (PAR), sertraline (SER) and its main metabolite desmethylsertraline (Nor-SER); the anti-inflammatories and/or analgesics and antipyretics, further referred only as anti-inflammatories, diclofenac (DIC) and its main metabolite 4-hydroxydiclofenac (4-OH-DIC), ibuprofen (IBU), naproxen (NAP), paracetamol (PARA) and its transformation product 4-aminophenol (4-PARA); and the hormones 17β-estradiol (E2) and its main metabolite estrone (E1) and 17α-ethinylestradiol (EE2; Table 1).

The pharmaceuticals in study, key representatives of major classes of pharmaceuticals, were selected based on the EU watch list, their high consumption, pharmacokinetics, physicochemical properties, persistence, previous studies on the occurrence on the aquatic environment, and their potential toxicological impact, both on humans and on the aquatic environment [10,11,12,13,14]. This review will provide a more realistic water quality assessment contributing for a more integrative approach to rank and prioritize pharmaceuticals, based on an integrated assessment of ERA and exposure in the aquatic environment.

“Water is not a commercial product like any other but, rather, a heritage which must be protected, defended and treated as such”, the claim by the EU WFD contrasts with the poor ecological status of European freshwater bodies, where only 43% achieve a good ecological status. In addition, and despite the enormous efforts, the picture that emerges regarding ecological status is still incomplete, fragmented and with contradictory assessments of the situation. Therefore, it is important to obtain a better understanding of the regional and global context, concerning the environmental risk posed by pharmaceuticals in the aquatic environment.

## 2. Toxicity

Since pharmaceuticals are continuously introduced into the aquatic environment, they can promote toxic effects on living organisms, even when present at concentrations on the ng L^−1^ level [15]. This potential for negative effects of pharmaceuticals even at sublethal concentrations, namely for aquatic organisms, has been of concern since the issue was first brought to attention in 1985 [2]. Therefore, their presence poses a threat to the quality of water resources [5,16].

Pharmaceuticals have a relatively clear mode of action in target organisms, and given that fish and invertebrates share more drug targets with humans, it would be expected that they would also respond to pharmaceuticals in a similar way. However, when non-target-species are exposed, unknown effects and potential risks need to be assessed. One example is the impact of EE2 in the feminization of fish [17,18,19]. Nonetheless, all the ecotoxicological risks associated to the ubiquitous occurrence of pharmaceuticals in aquatic ecosystems are far from known [3].

Sorption to sediments is one factor that influences toxicity of pharmaceuticals, although higher sorption to sediments results in an apparent reduction of bioavailability and toxicity, the activity of benthic invertebrate in sediments results in a higher exposure for these organisms [20].

Moreover, bioaccumulation and biomagnification should also be accounted for since they can increase toxicity [17]. These parameters are also related to log D_ow,_ since compounds with values higher than 3 have a tendency for bioaccumulation [17,21], which means that the ionization state can influence the toxicity of pharmaceuticals, and that the pH variability in surface water should also be taken into account [17].

A bibliographic search of the scientific literature was conducted on Google Scholar using the search terms “ecotoxicology” and each of the selected compounds. All the publications that presented ecotoxicological studies on the selected compounds, referring to the concentrations, were included. Below, the ecotoxicological data in the aquatic biota was reviewed, presenting the toxicity data obtained from 120 exposure studies of three trophic levels of non-target organisms, algae (Figure 1), invertebrates (Figure 2) and fish (Figure 3), Appendix A. The data was divided by the different endpoints found in the literature: no observed effect concentrations (NOEC), lowest observed effect concentrations (LOEC), effective concentration (EC50) and lethal concentration (LC50). These endpoints are expected to have increasing concentrations, since they were organized from the more susceptible endpoint (NOEC) to the less one (LC50). However, each endpoint encloses various species of the same trophic level and different toxicological tests like immobilization, growth, luminescence, reproduction, morphology, behavior, etc. When no experimental data was available, L(E)C50 values were estimated with ECOSAR 2.0. This program estimates data on acute toxicity through the molecule structure, sometimes underestimating toxic effects. The data was also divided in acute and chronic toxicity, depending on the time of exposure and trophic level. For algae, acute toxicity was considered when the toxicity tests lasted until 4 days (96 h), longer exposures were considered chronic toxicity. Regarding invertebrates, with the exception of *Brachionus calyciflorus* (were 2 days was considered chronic data, since it has a shorter life cycle), acute toxicity was accounted when the exposure took place until 2 days (48 h) and chronic toxicity when it was equal or longer than 7 days. For fish, tests until 4 days (96 h) were included in acute toxicity data and exposures equal or above 7 days entered the chronic toxicity data. These criteria were based on OECD tests for each trophic level [22].

Although, as expected, some therapeutic groups presented higher toxicity, such as hormones, which can promote endocrine modifications, all therapeutic groups presented toxicity at low concentrations, highlighting the ecotoxicity of the selected pharmaceuticals [138]. Overall, considering all trophic levels, all therapeutic groups with the exception of anxiolytics, had at least one toxicity report for concentrations below 1 µg L^−1^, near the concentrations found in the aquatic environment.

Considering the toxicity of the selected pharmaceuticals in all trophic levels, we could observe that the most sensitive one, with the lowest concentrations promoting toxic effects was fish, followed by invertebrates and algae. The limitation of this analysis is that, regarding fish, there were also toxicity data obtained through cell line or tissue testing, which can be difficult to extrapolate to the entire organism. The therapeutic group with higher toxicity, mainly chronic toxicity in fish and invertebrates, are hormones. Additionally, the pharmaceutical that presented higher toxicity, with the lowest concentration promoting toxic effects, was EE2 at 0.1 ng L^−1^ in fish (NOEC, chronic toxicity) [123]. The highest concentrations promoting toxicity were detected in fish (LC50, acute toxicity), for CLA, CIP and ERY (1 g L^−1^), [23,123,126].

Ecotoxicological chronic studies on pharmaceuticals are lacking, meaning that many questions about the threat to the environment of pharmaceuticals remain unanswered. Additionally, the actual exposure scenario regards multiple pharmaceuticals, posing uncertainty regarding toxicology in long-term exposure. If many pharmaceuticals are present and share the same mode of action, then the toxicity of this mixture could be higher than if only one pharmaceutical is present, being usually considered the concept of concentration addition, although antagonistic and synergistic effects may also occur. This could result in risk underestimation, as the typical exposure is toward multicomponent chemicals [139,140,141,142].

One example of mixture effects was observed when using a mixture of anti-inflammatories (DIC, IBU and NAP). In this case, the acute toxicity was detected at concentrations where little or no effect was observed for the chemicals individually [20]. Even in mixtures with pharmaceuticals belonging to different therapeutic groups, additive and synergistic effects were reported. A mixture with E2 and FLU promoted a decrease in the reproductive success of *D. magna* more significantly than either chemical compounds alone [143]. Another example was provided by exposing *D. magna* to a mixture of CAR and a lipid lowering agent, which exhibited stronger effects during immobilization tests than the single compounds at the same concentration [20].

Taking into account mixture effects, some research has already been developed focusing on toxic effects, and not on specific pharmaceuticals. This was already used to evaluate wastewater treatment plants (WWTPs) removal efficiencies, by evaluating and comparing the toxicity (androgenecity, cytotoxicity, anti-estrogenicity and *L. variegatus* decrease in reproduction and biomass) both in wastewater influents (WWIs) and efluents (WWEs) [144,145].

Additionally to the active compounds of pharmaceuticals, excipients and additives are also present in medicines, that may contain endocrine disrupting chemical excipients and additives [138].

The measured concentrations of some of the selected pharmaceuticals reported for surface water all over the world surpassed the concentrations here described for toxicity, which suggests that the aquatic biota could be vulnerable to the presence of pharmaceuticals in their environment, and that toxic effects are expected to occur with unexpected outcomes [146].

It is unlikely that pharmaceuticals present in drinking water may pose a risk to the human health through chronic exposure, however, the toxicological implications are not clear [147]. Furthermore, studies have shown that infants may have difficulty in metabolizing drugs therefore, being more vulnerable to the toxic effects of these compounds [17].

As referred, many pharmaceuticals have the potential for bioaccumulation and biomagnification, and chronic effects on ecosystems cannot be ignored for animals at the higher end of the food web [148]. Thus, the health hazard of human exposure by ingestion of contaminated foods should also be taken into account [17].

### 2.1. Anxiolytics

No ecotoxicological data was found in literature for ALP, LOR and ZOL, and for that reason, all the results for this therapeutic group were obtained from ECOSAR 2.0 [18]. In decreasing order, the more toxic was ZOL, followed by ALP and LOR. The trophic level with the lowest reported concentrations producing toxicity was algae (from 0.211 to 6.07 mg L^−1),^ followed by fish (from 0.248 to 43.1 mg L^−1^) and invertebrates (from 1.55 to 39.4 mg L^−1^).

### 2.2. Antibiotics

Observing the acute toxicity for antibiotics, since there is little data on chronic endpoints, the pattern for the three trophic levels was similar for all antibiotics, with algae being more susceptible at lower concentrations (from 0.0018 to 20.6 mg L^−1^), followed by invertebrates (from 0.22 to 120 mg L^−1^) and fish (from 84 to 1000 mg L^−1^). If we compare each antibiotic, concerning invertebrates, it can be observed that CLA and CIP presented similar results, but when compared with ERY, lower concentrations (220 µg L^−1^) of this antibiotic can produce the same toxic effects, in this case growth inhibition [23].

In this therapeutic class, in addition to direct toxicological risks, concern has been raised about the potential for the antibiotic residues in water, since they are typically found in the aquatic environment at subtherapeutic concentrations, promoting the emergence of resistant bacteria and subsequent development of more resistant and virulent pathogens [149]. These bacterial resistances, through horizontal gene transfer, may end up in human pathogens, raising questions on human health and the stability of the ecosystem [150,151,152,153,154].

This emergence of bacterial resistance presents one of the major emerging threats to human health and is by far the highest risk for humans of having medicinal products residues in the environment [155]. Furthermore, historical evidence appears to indicate that in the aquatic environment resistance might be acquired faster than in the terrestrial environment [156].

Corroborating the effects on bacteria, changes in biomass and growth rate were reported at concentrations above 5.7 µg L^−1^ [47]. This therapeutic class can also induce immunotoxicity in the freshwater mussel at low concentrations, between 2 ng L^−1^ and 1100 ng L^−1^ [157].

### 2.3. Lipid Regulators

In this group, the pattern observed with both previous therapeutic groups was not so clear, with median concentrations similar in all trophic levels for acute toxicity. Observing these data, SIM was clearly the pharmaceutical, which promoted toxicity at lower concentrations for invertebrates (160 ng L^−1^) and fish (765 µg L^−1^) [26,56]. However, data on chronic toxicity, only available for GEM in two trophic levels, showed that the highest toxicity regarded fish (1.5 µg L^−1^), followed by invertebrates (78.0 µg L^−1^) [51,85].

### 2.4. Antiepileptics

For CAR, once again, the pattern of acute data, was similar to that registered for anxiolytics and antibiotics, with the lowest concentrations promoting toxicity at 10.0 µg L^−1^, 20 000 µg L^−1^ and 0.01 µg L^−1^ for algae, fish and invertebrates, respectively [27,44,57]. Considering the chronic data, similar concentrations were found to produce toxicity in invertebrates and fish trophic levels, ranging from 25 to 25,000 µg L^−1^ [27,29].

### 2.5. SSRIs

This therapeutic group has the peculiarity that the phylogenetically ancient and highly conserved neurotransmitter and neurohormone serotonin has been found in invertebrates and vertebrates, although its specific physiological role and mode of action is unknown for many species [158]. Many biological functions within invertebrates, such as reproduction, metabolism, molting and behavior, are under the control of serotonin [159]. Therefore, the pharmaceuticals in this therapeutic group could have tremendous effects on these and other organisms [160]. These facts are in agreement with those found in acute toxicity data found, since for all trophic levels this group had globally the lowest concentrations, which promoted toxic effects, being some of these on reproduction, survival and behavior [161].

When observing these data, the most sensitive trophic level was the invertebrates (0.1 µg L^−1^), followed by algae (12.1 µg L^−1^) and fish (72.0 µg L^−1^) [33,57,92]. In invertebrates, the pharmaceuticals with higher toxicity were FLU (100 ng L^−1^) and its metabolite Nor-FLU (300 µg L^−1^) and SER (100 µg L^−1^). On the other side, PAR was the one with lower toxicity [33,36,57,69]. In algae, the pharmaceutical with highest toxicity was SER, however, in invertebrates, FLU surpassed SER toxicity.

The only metabolite referred in the literature concerning toxicity studies was Nor-FLU, with data for algae and invertebrates. When comparing with FLU (algae and invertebrates), it is clear that the median concentrations inducing toxicity were always lower [162].

Studies performed on SER and FLU demonstrated the influence of pH on toxicity, since the uncharged drug can pass easier through the membrane and act inside the cells, showing a tenfold increased toxicity when shifting the pH closest to their pKa, increasing the nonionized form, from 6.5 to 8.5 and from 7.8 to 9, respectively [17,89,92].

### 2.6. Anti-Inflammatories

Most anti-inflammatories induce the nonspecific inhibition of prostaglandins. This, in turn, means that there is the potential for effects on any of the normal physiological functions mediated by prostaglandins. In fish, for instance, prostaglandins influence mechanisms of behavior and reproduction and, therefore, they can act as endocrine disruptors or modulators, because they can exert their effects by mimicking or antagonizing the effects of hormones, alter their pattern of synthesis and metabolism and modify hormone receptor levels, leading to possible adverse effects [7,163,164,165]. However, different and unexpected toxicity effects were also observed. One of the first was reported in Pakistan, where a catastrophic decline in the Oriental White-backed Vulture population (95%) originated from the exposure to DIC contaminated live-stock carcasses, which promoted fatal renal disease [98,138].

Overall, excepting anxiolytics, anti-inflammatories were less toxic than the other therapeutic groups. Regarding the lowest concentrations that produced acute toxicity in the three trophic levels, invertebrates had the lowest value (10 ng L^−1^), followed by algae (10 µg L^−1^) and fish (90 µg L^−1^), however, when using median values, the differences become less clear [37,44,57,96]. As for chronic data, higher toxicity was observed in fish (500 ng L^−1^) and invertebrates (200 µg L^−1^), when compared with algae (4.01 mg L^−1^), which is in line with the already referred anti-inflammatories mode of action [70,95].

Data for each anti-inflammatory showed no clear pattern, nonetheless, except for invertebrates, NAP and PARA seemed to have lower toxicity than DIC and IBU. When performing a comparison between DIC and its metabolite (4-OH-DIC) in invertebrates and fish, one could observe that they have similar toxicities. Conversely, PARA transformation product (4-PARA) presented higher toxicity than the parent molecule in all three trophic levels.

### 2.7. Hormones

Although hormones like E1, E2 and EE2 are mainly used for contraception purposes, the physiological effects are not restricted to effects on reproductive and sexual development, and can target mitochondrial function, energy metabolism and cell cycle control [165].

For acute toxicity, there is only data on algae and invertebrates, and algae presented higher toxicity since the lowest concentration promoting toxic effects was at 162 µg L^−1^, lower than the 1500 µg L^−1^ observed in invertebrates [43,78]. Nonetheless, the toxicity promoted by this therapeutic group is mainly expected to be detected through chronic toxicity, however, these data could only be obtained for invertebrates and fish. Considering chronic data, in these two trophic levels, hormones presented higher toxicity than the other therapeutic groups, since the lowest concentrations reported were of 100 ng L^−1^ and 0.1 ng L^−1^, for invertebrates and fish, respectively [81,82,123]. It should also be noted that, the highest concentration found that promoted toxicity for fish was also very low (1188 ng L^−1^) [102].

Individually, there were no differences observed between E1 and E2 toxicity, while EE2 seems the most toxic compound regarding chronic toxicity in invertebrates and especially in fish, where the 36 results available presented concentrations below 44 ng L^−1^ [135]. Namely, when two different fish species were exposed to EE2 at 3 ng L^−1^ and 4 ng L^−1^ they suffered sex gender reversal, from male to female, which can strongly unbalance the aquatic ecosystem [130,136,138].

## 3. Environmental Risk Assessment

The data regarding occurrence and toxicity already presented is crucial in order to perform the ERA, and can be used to select the pharmaceuticals that are more prone to induce toxic effects in aquatic biota [166]. The risk assessment, mentioned in the EMA guideline on the ERA of medicinal products for human use [4], is performed through the risk quotient (RQ) calculation, dividing the PEC by the PNEC for each pharmaceutical, observing three different trophic levels (algae, invertebrates and fish). If RQ is equal or above 1, there is a potential environmental risk situation, whereas when values are lower than 1, no risk is expected. However, a certain risk could be expected for the substances with a RQ between 0.1 and 1 [167,168]. However, this guideline is only applied for marketing authorizations and for pharmaceuticals marketed after 2006. Additionally, it does not constitute a valid criterion upon which to base the refusal of a market authorization of medicinal products for human use in the EU [6]. Our evaluation of the potential ecotoxicological risk posed for the aquatic compartment was based on a dual approach: one using the worst case scenario, as stated by the EMA guideline on the ERA [4], where the maximum individual concentrations of pharmaceuticals found in the respective aquatic compartment were used as MEC [15,169,170], and another using the median concentrations for each pharmaceutical as MEC [171]. This evaluation can also be an important tool to suggest the inclusion or removal of pharmaceuticals in the watch list of the Directive 2013/39/EU.

As discussed, some concentrations compiled in surface water are higher than the levels that induce toxicity, not applying any uncertainty factor (UF) for the PNEC calculation. Additionally, some studies have indicated that concentrations of several pharmaceuticals belonging to different therapeutic groups can promote toxic effects on negatively impacted aquatic biota, presenting RQ higher than 1 [11,18,139,157,172,173].

As referred, aquatic biota inhabiting the receiving environment are unintentionally exposed throughout their lifetime to a complex mixture of residual pharmaceuticals and these mixtures can exhibit a greater effect than individual compounds [20,174,175]. Therefore, it is a challenge to address the concerns related to the chronic effect, low-level exposure to these compounds, including exposure of sensitive subpopulations to pharmaceutical mixtures [17,174].

### 3.1. Predicted No-Effect Concentration (PNECs)

Based on the toxicity data (Figure 4, Figure 5, Figure 6 and Figure 7), Table 2 presents the PNECs for the selected pharmaceuticals. These values were calculated by applying an UF of 100 and 10 to the long-term EC50 and NOEC values, and an UF of 50 and 1000 to the short-term LOEC and L(E)C50 values, respectively, available in the literature. The UF is an expression of the degree of uncertainty in the extrapolation from the test data on a limited number of species to the actual environment [4]. As referred, when no experimental data are available, L(E)C50 values were estimated through ECOSAR 2.0.

It should be taken into account that the choice of toxicity data can obviously affect the outcome [15]. However, the results obtained for the PNECs were directly related to the toxicity data, and a similar pattern to the toxicity data was observed, with the therapeutic groups and pharmaceuticals with higher toxicity presenting the lowest PNEC values.

### 3.2. Risk Assessment

Using the occurrence data obtained from Part I of this review (Appendix A) and the PNECs previously calculated (Table 2), RQs were deemed for all the selected pharmaceuticals in the different aquatic compartments and are presented in Figure 4, Figure 5, Figure 6 and Figure 7 [146].

In general, the results revealed that RQs higher than 1 could be observed for all the aquatic bodies, posing ecotoxicological pressure in all of these compartments.

#### 3.2.1. Wastewater Influents

The RQs observed in WWIs were the highest from all the aquatic compartments, as well as the concentrations of the selected pharmaceuticals (Figure 4). The highest value (274,816) and median (13,400) were observed for IBU in invertebrates. Anti-inflammatories were the therapeutic group with the highest RQs median, both for the maximum and median values, followed by antiepileptic and hormones.

With the exception of anxiolytics, all the other therapeutic groups presented RQs > 1 for at least two trophic levels, being SSRIs and anxiolytics the only groups that did not present risk RQs > 1 for all pharmaceuticals.

For the anxiolytics and antibiotics, the algae were clearly the most susceptible trophic level, presenting higher RQs. Another clear pattern was observed for the hormones, where for fish all the RQs (median and maximum) were higher than 1 and with the maximum values between 129 and 17 271

These results demonstrated that the concentrations reaching the WWTPs could clearly endanger all the trophic levels that might be exposed to this aquatic matrix.

#### 3.2.2. Wastewater Effluents

This aquatic compartment presents lower RQs than the WWIs (Figure 5). Antibiotics along with antiepileptic, anti-inflammatories and hormones were the therapeutic groups with highest RQs values. When considering median values alone, antibiotics have lower RQs than the other three therapeutic groups. The values observed for antibiotics can promote an even bigger problem than the direct toxicity to aquatic organism: the emergence of bacterial resistance. Nonetheless, the ERA approach does not address this issue [155].

Anxiolytics continued to present RQs values lower than 1. Antibiotics, on the other hand, had the highest value observed for CIP in algae (100,258). As in WWI, antiepileptic continued to present the highest median RQs.

In WWI, some metabolites of the SSRIs presented similar or slightly higher RQs than the parent compounds (N-CIT and Nor-SER), in WWE a similar pattern was observed. These results highlight the fact that parent compounds and metabolites reached WWEs, and that the concentrations found in this matrix were able to promote toxic effects in the aquatic biota. This fact suggests that metabolites and transformation products should also be monitored in the environment.

Regarding the anti-inflammatories therapeutic group, DIC and IBU stand out from the other pharmaceuticals presenting clearly higher median and maximum RQs. As for hormones and observing the fish trophic level, as in the WWI, all RQs were higher than one.

#### 3.2.3. Surface Waters

This aquatic body was clearly more problematic to the environment when compared to WWIs and WWEs since here is where most of aquatic life inhabits. However, like in the previous water compartments, RQs higher than 1 were observed for all trophic levels and therapeutic groups, with the exception of the anxiolytics. Antiepileptic, anti-inflammatories, antibiotics and hormones remained the therapeutic groups with the highest RQs (Figure 6).

The highest maximum and median values regarded IBU (16,327) and CAR (138) in invertebrates. There were still eighteen pharmaceuticals (AZI, CLA, CIP, ERY, BEZ, GEM, SIM, CAR, CIT, FLU, SER, DIC, IBU, PARA, E1, E2, αE2 and EE2) with RQs above 1.

Antibiotics still presented all of their maximum RQs higher than 1 for algae, whereas lipid regulators presented the same pattern for invertebrates. As for the antiepileptic, their maximum values were above one for algae and invertebrates. In SSRIs therapeutic group, CIT, FLU and SER were the ones that presented RQs higher than 1, contributing to the possible risk posed by this group. The hormones, regarding fish, still presented all median RQs higher than 1, with the exception of E1, with the EE2 obtaining the highest RQs for this trophic level.

As already mentioned, this water body, encompassing rivers and lakes, should be free from risk. Nonetheless, from the 28 detected pharmaceuticals, 18 presented maximum RQs above 1 and even 8 had median RQs superior to 1, posing a threat to all the aquatic organisms.

#### 3.2.4. Other Water Bodies

The results obtained for the RQs of other water bodies are presented in Figure 7. The RQs, in decreasing order, were groundwater, seawater, mineral water and drinking water. Since drinking water is usually obtained through surface and groundwaters, these results suggest that the sources used to produce drinking water were the ones with lower pharmaceutical contaminations or that water treatment plants were removing the selected pharmaceuticals. The results obtained in seawater can be biased, since fresh water organisms were used to evaluate the risk and there are reports that marine organism can be more vulnerable, increasing the risk in this water matrix [178,179].

All of these compartments presented values above 1 for all trophic levels, with the exception of mineral water in algae, highlighting, once again, the pressure sustained by the aquatic organisms in all the aquatic compartments. This indicates that there is also a possible risk for humans. The therapeutic groups with higher maximum and median RQs are hormones, antiepileptic, antibiotics and anti-inflammatories, being the same groups as in WWIs.

With the exception of anxiolytics, all therapeutic groups still present RQ medians above 1, with 12 pharmaceuticals with maximum values above 1, and 11 with medians also higher than 1. If we use the threshold of 0.1, where some risk might be expected, we can find 16 of these compounds. Some RQs are still extremely high, with EE2 presenting values up to 91,150 and a median of 6091 for fish in groundwater. The higher RQs observed in these compartments were for CAR and IBU in invertebrates, and for hormones in fish.

Viewing these results, it is possible to observe not only high risk for aquatic organism in wastewaters but, despite the RQ reduction, several pharmaceuticals still promote risk in other supposedly cleaner aquatic matrices. Additionally, additive or even synergistic effects can occur, especially in the pharmaceuticals with the same mechanism of action [139,140,141,142].

### 3.3. Mitigation Measures

The RQs obtained for all the water compartments, particularly the RQs higher than 1 in surface waters and in the other water bodies, raise not only the issue of toxicity for the aquatic environment but also for humans using these aquatic bodies as a source of drinking water and also for whom eating animals living in these contaminated environments. Therefore, mitigation measures should be implemented to prevent high RQs in these important water resources. These measures should begin through the awareness of the problem. For example, in Sweden, an environmental classification system for drugs has been established through collaboration between producers, authorities and the public health care. This system assesses the environmental risk of pharmaceuticals being publicly available, therefore, the market can demand for medicines with less environmental impact, stimulating producers to design more environmentally friendly medicines [17]. This includes the concept of green pharmacy, where the design of pharmaceutical products focus also on their high metabolization and environmental degradation, reducing the environmental burden and improving environmental safety and health impacts [138].

Additionally, the ERA guideline on human pharmaceuticals should be revised in order to: enter the risk benefit analysis; impose its revision every five years with the new available data; incorporate the pharmaceuticals marketed before 2006 and include metabolites and transformation products [6].

Another issue already referred is the possible improvement of WWTPs removal efficiencies and the improvement of toxicity studies. The main challenges regarding the assessment of ecotoxicity are the scarce information available for some of the selected pharmaceuticals, namely chronic data and effects on multiple life stages or even multiple generations, which mimics the environmental exposure. Behavior studies are also lacking, before death and other major toxicity effects there can be diminished sexual interest, fear or activity, increased curiosity, etc. Although this seems like small behavior changes, it can be enough to unbalance an ecosystem eliminating one or more species by being unable to escape from predators or by lack of reproduction. This leads to another point that ecotoxicity studies should also be performed in ecosystems, because some effects previously referred are more evident in this type of studies. It should also be considered the increase of species tested and mixture effects, not only between different pharmaceuticals of the same therapeutic group but also from different groups and with other substances, like heavy metals. Given the pharmaceutical environmental presence in mixtures and with other substances, additive or even synergistic effects occur; therefore, the real hazard may be greater than that calculated [24,33,180,181]. Another issue that is neglected with the traditional ecotoxicity studies is the emergence of bacterial resistance. We are testing antibiotics in algae, invertebrates and fish but probably the biggest impact will be on bacteria, namely on the emergence of resistant bacteria that can reach not only aquatic animals but also humans. Additionally, the emergence of bacterial resistance is a major concern involving the presence of pharmaceuticals in the aquatic compartment, which is more prone in this environment [6,182].

This could generate benefits in water resource management, by providing the means for cross-compliance measures in environmental regulation and providing an adequate risk assessment for pharmaceuticals mixtures [183]. In this way, the complete scenario of the contamination of pharmaceuticals in the aquatic environment and their risk could be performed, contributing to future improvements in minimization measures and legislation.

## 4. Final Remarks

A literature review was conducted in order to understand the toxicity and ERA of pharmaceuticals in the aquatic environment. In this context, a broad and specialized background was obtained, enabling an overview of the state of the art in these subjects.

Regarding the toxicity data, although the differences observed between different therapeutic groups and within each therapeutic group, all therapeutic groups with the exception of anxiolytics, had at least one toxicity report for concentrations below 1 µg L^−1^. The trophic level with the lowest concentrations promoting toxic effects was fish, followed by invertebrates and algae, emphasizing that fish, the trophic level closer to humans, are more prone to toxicity effects from the selected pharmaceuticals.

The results also show that pharmaceuticals with higher RQs are not the ones with higher occurrence and that proper toxicity data is important to a correct evaluation of the ERA.

The ERA performed for the pharmaceuticals in the different aquatic compartments revealed, with the exception of anxiolytics, RQs higher than 1, not only for WWIs but also for all aquatic compartments, for all trophic levels and therapeutic groups.

The therapeutic groups with the highest RQs in all aquatic compartments are hormones, antiepileptics, anti-inflammatories and antibiotics and the pharmaceuticals with the highest values are the EE2, CAR, IBU, CIP and AZI in all aquatic compartments. Highlighting threat to all the aquatic organisms exposed, namely on the feminization of fish by EE2 and its impact on aquatic ecosystems. Additionally, two antibiotics are among the pharmaceuticals with higher RQs and the ERA does not evaluate the emergence of bacterial resistance. If this issue was also evaluated it would probably confirm why they are considered the therapeutic group with the highest risk for humans, regarding the residues of medicinal products in the environment.

Unfortunately, the pressure of pharmaceuticals on aquatic bodies will continue to rise, and, therefore, mitigation measures and changes in legislation must be implemented.

## Figures and Tables

**Figure 1 molecules-25-01796-f001:**
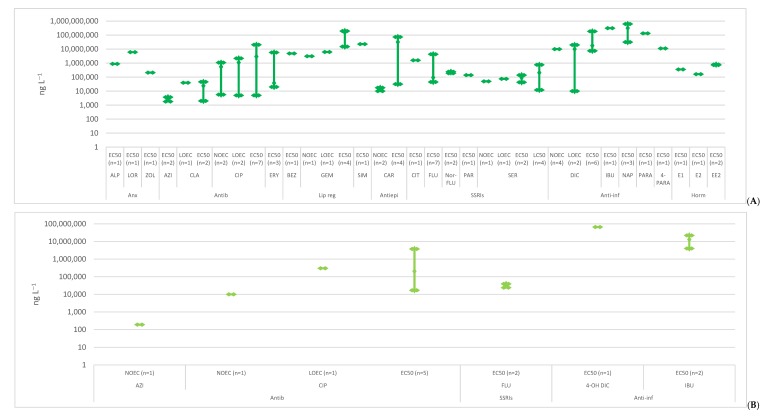
Median, maximum and minimum concentration values reported for acute (**A**) and chronic (**B**) toxicity concerning algae. (Anx—anxiolytics; Antib—antibiotics; Lip reg—lipid regulators; Antiepi—antiepileptics; SSRIs—Selective serotonin reuptake inhibitors; Anti-inf—anti-inflammatories; Horm—hormones) [23,24,25,26,27,28,29,30,31,32,33,34,35,36,37,38,39,40,41,42,43,44,45,46,47,48,49,50,51,52].

**Figure 2 molecules-25-01796-f002:**
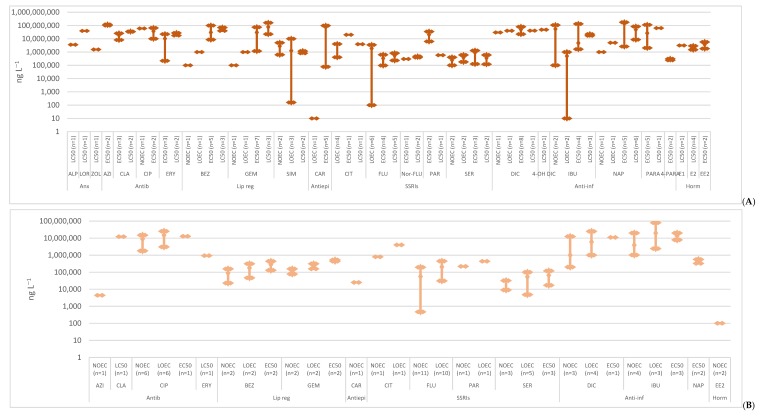
Median, maximum and minimum concentration values reported for acute (**A**) and chronic (**B**) toxicity data concerning invertebrates. (Anx—anxiolytics; Antib—antibiotics; Lip reg—lipid regulators; Antiepi—antiepileptics; SSRIs—Selective serotonin reuptake inhibitors; Anti-inf—anti-inflammatories; Horm—hormones) [23,27,28,29,30,31,32,35,36,40,41,42,44,46,51,52,53,54,55,56,57,58,59,60,61,62,63,64,65,66,67,68,69,70,71,72,73,74,75,76,77,78,79,80,81,82,83,84].

**Figure 3 molecules-25-01796-f003:**
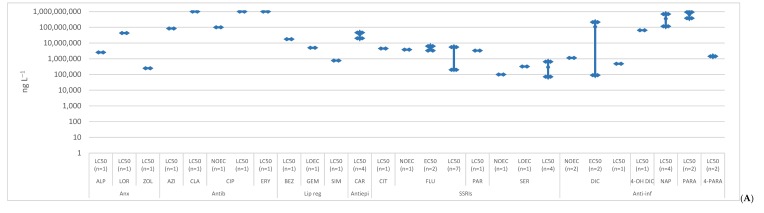
Median, maximum and minimum concentration values reported for acute (**A**) and chronic (**B**) toxicity data concerning fish. (Anx—anxiolytics; Antib—antibiotics; Lip reg—lipid regulators; Antiepi—antiepileptics; SSRIs—Selective serotonin reuptake inhibitors; Anti-inf—anti-inflammatories; Horm—hormones) [23,29,31,32,35,36,42,44,58,59,70,74,82,83,85,86,87,88,89,90,91,92,93,94,95,96,97,98,99,100,101,102,103,104,105,106,107,108,109,110,111,112,113,114,115,116,117,118,119,120,121,122,123,124,125,126,127,128,129,130,131,132,133,134,135,136,137].

**Figure 4 molecules-25-01796-f004:**
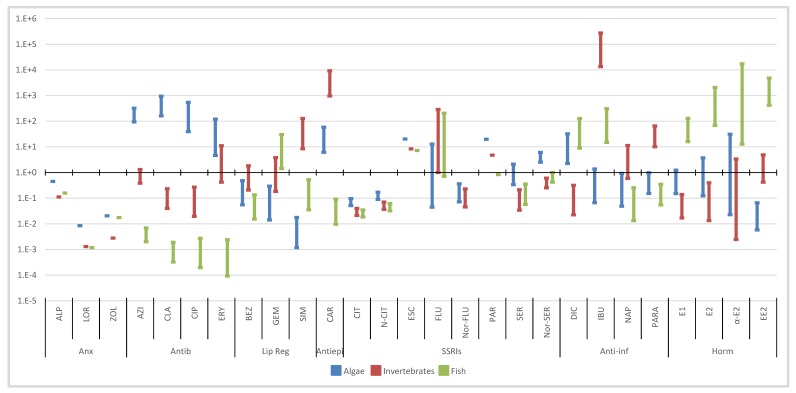
Median and maximum risk quotients of pharmaceuticals in WWIs for the three trophic levels. (Anx—anxiolytics; Antib—antibiotics; Lip reg—lipid regulators; Antiepi—antiepileptics; SSRIs—Selective serotonin reuptake inhibitors; Anti-inf—anti-inflammatories; Horm—hormones).

**Figure 5 molecules-25-01796-f005:**
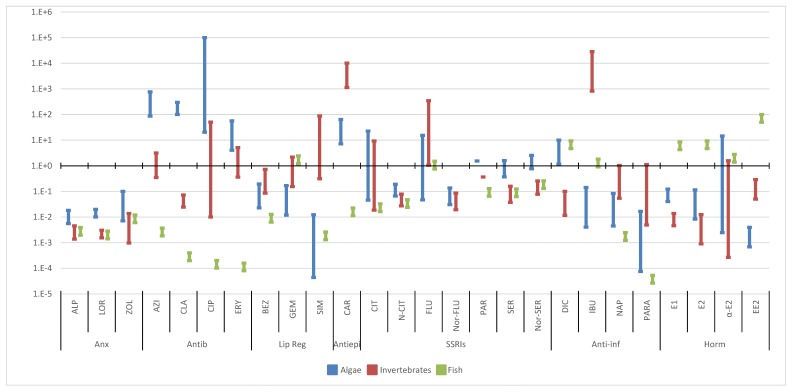
Median and maximum risk quotients of pharmaceuticals in WWEs for the three trophic levels. (Anx—anxiolytics; Antib—antibiotics; Lip reg—lipid regulators; Antiepi—antiepileptics; SSRIs—Selective serotonin reuptake inhibitors; Anti-inf—anti-inflammatories; Horm—hormones).

**Figure 6 molecules-25-01796-f006:**
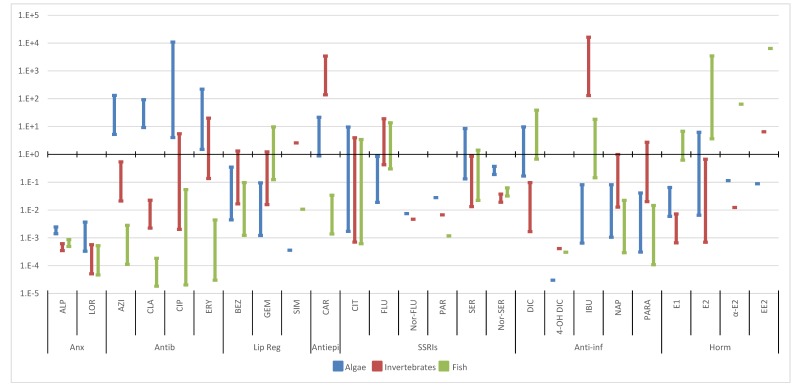
Median and maximum risk quotients of pharmaceuticals in surface waters for the three trophic levels. (Anx—anxiolytics; Antib—antibiotics; Lip reg—lipid regulators; Antiepi—antiepileptics; SSRIs—Selective serotonin reuptake inhibitors; Anti-inf—anti-inflammatories; Horm—hormones).

**Figure 7 molecules-25-01796-f007:**
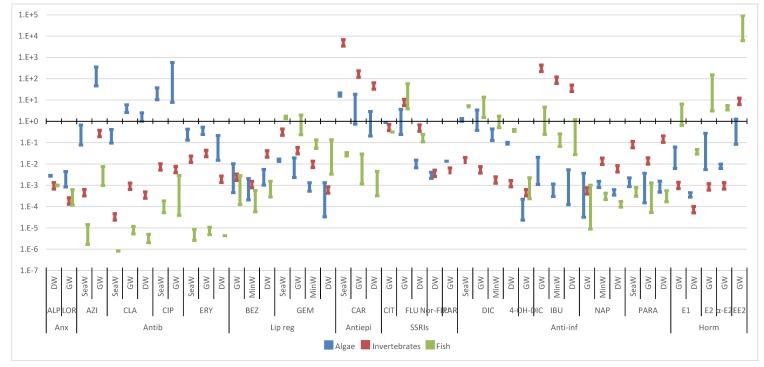
Median and maximum risk quotients of pharmaceuticals in other water bodies for the three trophic levels. (DW—drinking water; GW groundwater; SeaW—seawater; MinW—mineral water; Anx—anxiolytics; Antib—antibiotics; Lip reg—lipid regulators; Antiepi—antiepileptics; SSRIs—Selective serotonin reuptake inhibitors; Anti-inf—anti-inflammatories; Horm—hormones).

**Table 1 molecules-25-01796-t001:** Selected pharmaceuticals.

Therapeutic Group	Compound and Chemical Structure			
Anxiolytics (Anx)	Alprazolam (ALP)	Lorazepam (LOR)	Zolpidem (ZOL)	
	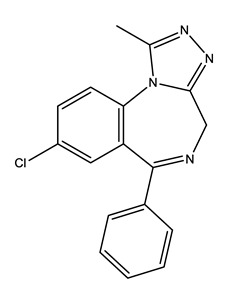	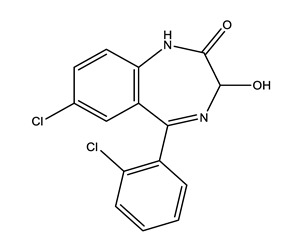	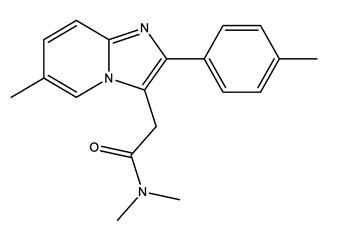	
Antibiotics (Antib)	Azithromycin (AZI)	Ciprofloxacin (CIP)	Clarithromycin (CLA)	Erythromycin (ERY)
	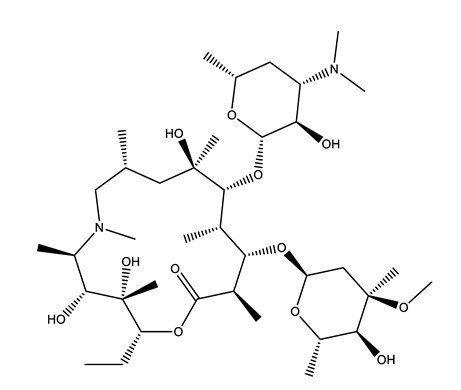	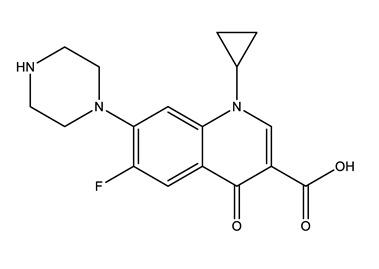	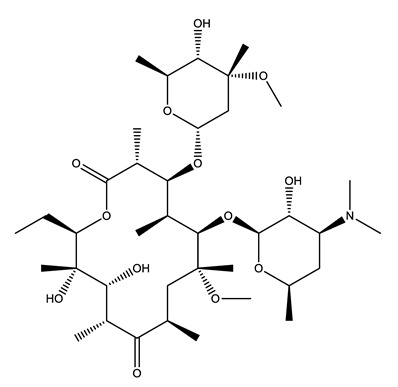	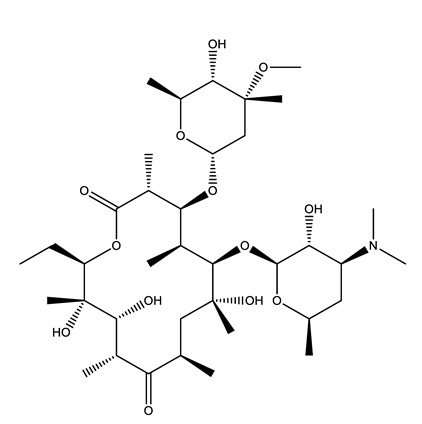
Lipid regulators (Lip Reg)	Bezafibrate (BEZ)	Gemfibrozil (GEM)	Simvastatin (SIM)	
	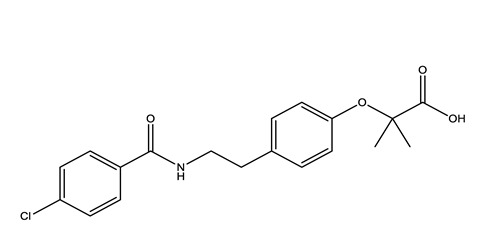	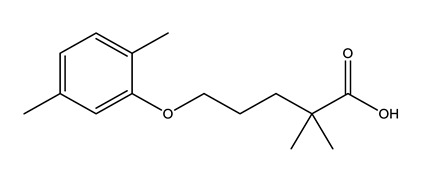	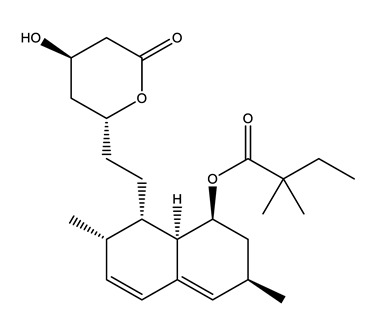	
Antiepileptic (Antiepi)	Carbamazepine (CAR)			
	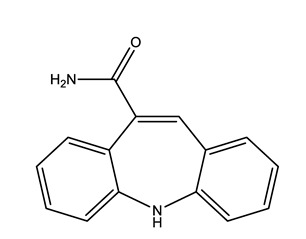			
Selective serotonin reuptake inhibitors (SSRIs)	Citalopram (CIT)	Desmethylcitalopram (N-Cit) (metabolite)	Escitalopram (ESC)	Fluoxetine (FLU)
	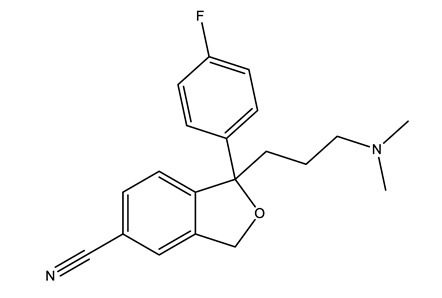	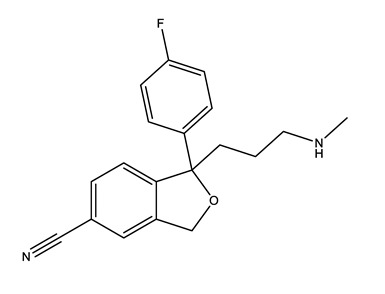	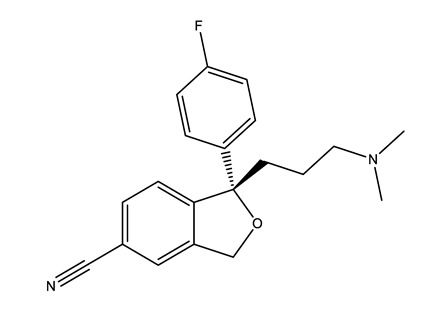	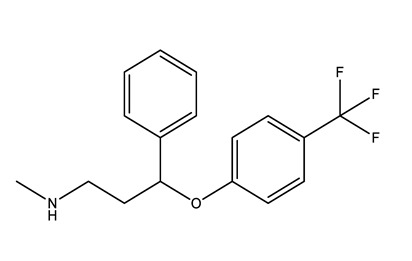
	Norfluoxetine (Nor-FLU) (metabolite)	Paroxetine (PAR)	Sertraline (SER)	Desmethylsertraline (Nor-SER) (metabolite)
	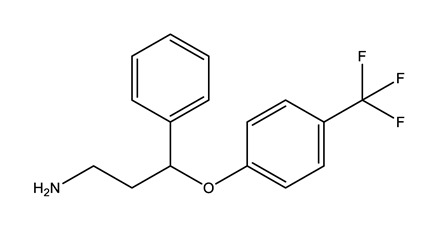	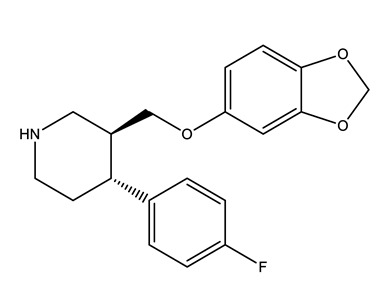	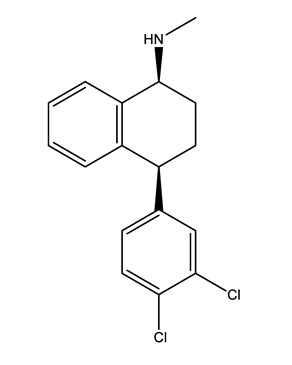	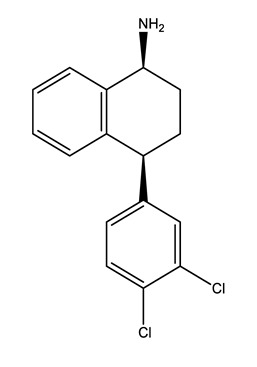
Anti-inflammatories (Anti-inf)	Diclofenac (DIC)	4-hydroxydiclofenac (4-OH-DIC) (metabolite)	Ibuprofen (IBU)	Naproxen (NAP)
	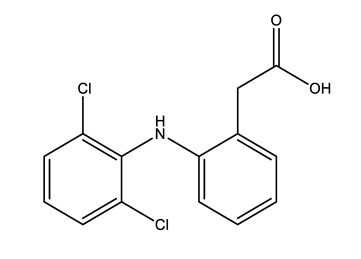	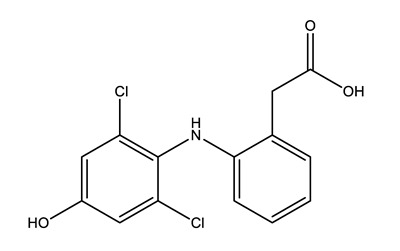	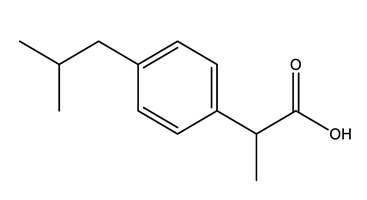	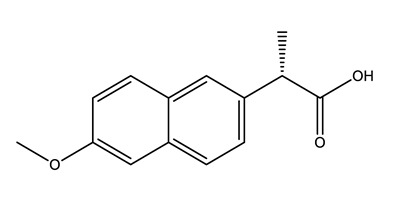
	Paracetamol (PARA)	4-aminophenol (4-PARA) (transformation product)		
	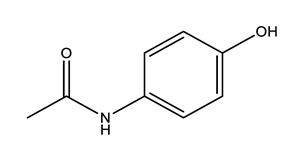	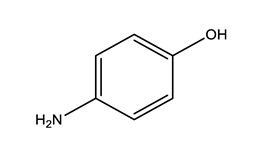		
Hormones (Horm)	Estrone (E1) (natural hormone/metabolite)	17β-estradiol (E2)	17α-ethinylestradiol (EE2)	
	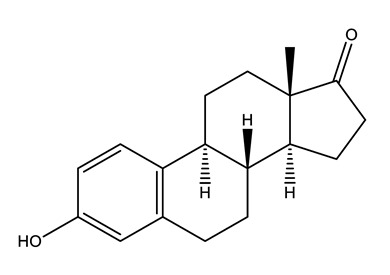	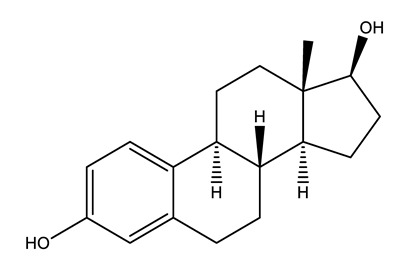	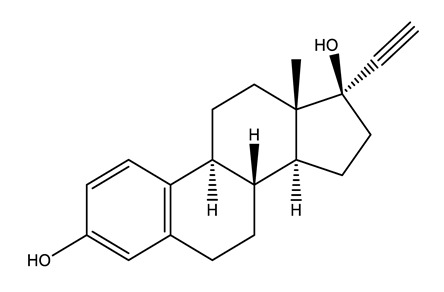	

**Table 2 molecules-25-01796-t002:** Predicted no-effect concentrations of the selected pharmaceuticals for algae, invertebrates and fish for the studied pharmaceuticals.

Therapeutic Group	Pharmaceutical	PNEC (ng L^−1^) Algae	PNEC (ng L^−1^) Invertebrates	PNEC (ng L^−1^) Fish
Anx	ALP	892 ^a,b^	3590 ^b,c^	2540 ^b,c^
	LOR	6070 ^a,b^	39,400 ^b,c^	43,100 ^b,c^
	ZOL	211 ^a,b^	1550 ^b,c^	248 ^b,c^
Antib	AZI	1.8 ^b^ [176]	440 ^e,g^ [83]	84,000 ^b^
	CLA	2 ^b^ [23]	8160 ^b^ [23]	1,000,000 ^b^ [23]
	CIP	5 ^b^ [35]	10,000 ^b^ [44]	1,000,000 ^b^ [44]
	ERY	20 ^b^ [23]	220 ^b^ [23]	1,000,000 ^b^ [23]
Lip reg	BEZ	4870 ^a,b^	1300 ^e,f^ [51]	17,600 ^b,c^
	GEM	15,190 ^b^ [51]	1180 ^b^ [53]	150 ^e,g^ [85]
	SIM	22,800 ^b^ [26]	3.2 ^d^ [56]	765 ^b,c^
Antiepi	CAR	31.6 ^b^ [27]	0.2 ^d^ [57]	20,000 ^b^ [44]
SSRIs	CIT	1600 ^b^ [30]	3900 ^b^ [58]	4470 ^b,c^
	FLU	44.99 ^b^ [33]	2 ^d^ [57]	2.8 ^e,g^ [177]
	Nor-FLU (M)	189 ^b^ [34]	300 ^b^ [61]	n.a
	PAR	140 ^b^ [30]	580 ^b^ [58]	3290 ^b,c^
	SER	12.10 ^b^ [33]	120 ^b^ [58]	72 ^b^ [92]
Anti-inf	DIC	200 ^d^ [37]	20,000 ^e,g^ [70]	50 ^e,g^ [95]
	4-OH-DIC (M)	660,300 ^e,f^ [38]	48,200 ^b,c^	65,200 ^b,c^
	IBU	40,100 ^e,f^ [39]	0.2 ^d^ [57]	180 ^e,g^
	NAP	31,820 ^b^ [40]	2620 ^b^ [53]	115,200 ^b^ [99]
	PARA	134,000 ^b^ [59]	2040 ^b^ [73]	378,000 ^b^ [42]
	4-PARA (TP)	11,300 ^a,b^	240 ^b^ [77]	1430 ^b^ [100]
Horm	E1 (NH/M)	355 ^a,b^	3160 ^b,c^	3.4 ^e,g^ [103]
	E2	162 ^a,b^	1500 ^b^ [78]	0.29 ^e,g^ [113]
	EE2	730 ^b^ [43]	10 ^e,g^ [81,82]	0.01 ^e,g^ [123]

M—metabolite; TP—transformation product; NH—natural hormone. ^a^: EC50 was estimated with ECOSAR. ^b^: UF = 1000. ^c^: LC50 was estimated with ECOSAR. ^d^: UF = 50 (uncertainty factor used for lowest observed effect concentrations (LOEC) and no observed effect concentrations (NOEC) in acute toxicity). ^e^: long-term data. ^f^: UF = 100. ^g^: UF = 10 (uncertainty factor used for LOEC and NOEC in chronic toxicity).

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
