# Peer review of "Selected Pharmaceuticals in Different Aquatic Compartments: Part II—Toxicity and Environmental Risk Assessment"

_molecules, 2020, doi:10.3390/molecules25081796_

Round 1
Reviewer 1 Report
1. Abstract. The third sentence from the end: "The therapeutic groups with the higher RQs...". Remove 2nd "and" at the end of the sentence.
2. Paracetamol is a drug with analgesic and antipyretic effects, without anti-inflammatory effect. So, anti-inflammatory group of drugs should be renamed. I suggest the following name for this group: Antiinflammatory/Analgesic-antipyretic
Author Response
- Abstract. The third sentence from the end: "The therapeutic groups with the higher RQs...". Remove 2nd "and" at the end of the sentence.
The authors agree with the reviewer suggestion and the manuscript was corrected accordingly.
- Paracetamol is a drug with analgesic and antipyretic effects, without anti-inflammatory effect. So, anti-inflammatory group of drugs should be renamed. I suggest the following name for this group: Antiinflammatory/Analgesic-antipyreti
The authors agree with the reviewer comment, therefore, they added a sentence describing the selected pharmaceuticals were this issue was clarified. This option avoids the increase of therapeutic group names throughout the manuscript, making the text easier to read.
Reviewer 2 Report
Very nice paper, very well written.
However, if possible, I would very much feel the need for the addition of a subchapter dedicated to the progress and main challenges regarding the methodology used for assessing the ecotoxicity.
Author Response
Very nice paper, very well written.
However, if possible, I would very much feel the need for the addition of a subchapter dedicated to the progress and main challenges regarding the methodology used for assessing the ecotoxicity.
The authors appreciate the reviewer comment and suggestion. The authors considered the inclusion of a subchapter on “main challenges regarding the methodology used for assessing the ecotoxicity”, but we did not want to increase the length of the manuscript, since it is already long. Additionally, there is enough data to write a review/opinion/short letter on this issue. However, the authors agreed with the development of this issue in the chapter “Mitigation measures”.
Reviewer 3 Report
This manuscript by Pereira et al is a review of the literature about toxicity data of 28 pharmaceutical compounds (active substances, metabolites and transformation products) and their respective environmental risk assessment. This review is intended to be complete, exhaustive but it is far from being the case, since many publications on this topic are missing. In this form, this review does not provide much more new information compared to other reviews on the same topic.
Major comments:
- As this is not a complete review of the literature, the authors should explain how they selected the publications. On which criteria? If authors deliberately removed publications, the criteria used to choose to remove or not a publication should be mentioned in the manuscript.
- Only one or two toxicity values are reported for a same species. For some compounds, it seems that these values are the lowest reported in the literature but this not the case for all the compounds. Explain the choice of the reported values.
- L.125: What is a low concentration? Because these toxic concentrations are still some order of magnitude higher than environmental concentrations (with some exceptions).
- Avoid sentences stating the obvious (e.g. L.129-130, L.136-140, …). L. 139 “once again, as anticipated, LC50 data presented the highest concentrations”. By definition, LC50 values will always be higher than NOEC and LOEC values. Otherwise there will be a problem.
- Some of the data for fishes correspond to data on cell lines or tissue. It is quite difficult to then extrapolate to the entire organism or even say that fishes are more sensitive to other organisms for which data were obtained on the entire organism.
- L141: What do the authors mean by "do not produce visible results". Maybe there is no chronic toxic effect of these compounds at all. Please rephrase.
- Figures 4-6: Risk quotients are classically calculated only using the most sensitive organisms, since by protecting the more sensitive obviously the other will be protected too. Thus, it is not necessary to calculate RQ for all the trophic levels. Otherwise, represent the data of toxicity as species sensitivity distribution and use hazard quotient 5%.
- Authors used data obtained on freshwater species to derive RQ for seawater whereas there are several publications dealing with marine organisms exposed to some of the selected compounds showing a strong sensitivity of marine organisms, much higher than freshwater organisms. And RQs below 1 have been calculated for antidepressants and antibiotics, among other compounds. The authors should modulate their text, or discuss this point.
- I do not see the point to perform Environmental/Ecological risk assessment for drinking waters. I would rather understand human health risk assessment.
- Concentrations measured in the different aquatic environments are reported in a previous publication, but it will be easier if the authors can provide the concentrations used here to calculate RQs, in the present publication. The reader will also be able to better understand how RQs (mean – max RQs) were calculated (what were the data used?).
- L463, 465: I do not agree. The author did not conduct a careful literature review or a complete overview of the state of the art, since too many publications are missing.
Minor comments:
- 21: Remove “and” at the end of the sentence.
- 47: Replace “aquatic mesocosm” by “aquatic ecosystem”
- Figures: Add the number of data used for each point (n= ?), and specify the meaning of the whiskers and the point (CI95% or min-max values? Means or medians?)
Sup data:
Be consistent throughout the table:
- Species names are needed, since for a same phylum different species can have different sensitivities.
- One line per species: all the data available for the same species should be written one after the other and not separated by the data of other species.
- EC50 (72h) data for algae species are considered by the authors either as acute or chronic ecotoxicity data. First line of Table S1 for Alprazolam, the authors considered EC50 (96h) for green algae as an acute ecotoxicity data. Ecotoxicity tests at 72h are often considered as acute toxicity tests but considering the generation time of algae species, it can also be considered as chronic ones. So, the authors should choose acute / or chronic, and be consistent throughout the table.
- Some tests are missing (e.g. p.43 last line – réf 84 or p.47 – réf 96, …).
Author Response
This manuscript by Pereira et al is a review of the literature about toxicity data of 28 pharmaceutical compounds (active substances, metabolites and transformation products) and their respective environmental risk assessment. This review is intended to be complete, exhaustive but it is far from being the case, since many publications on this topic are missing. In this form, this review does not provide much more new information compared to other reviews on the same topic.
The authors expect that, with the corrections performed, the reviewer point of view has changed. The authors believe that the way toxicity and environmental risk assessment is presented is different from other reviews, using medians and maximum values and providing data by trophic levels. Additionally, all the data is well organized and compiled in supplementary material, which allows other authors to easily use it for other research purposes.
Major comments:
- As this is not a complete review of the literature, the authors should explain how they selected the publications. On which criteria? If authors deliberately removed publications, the criteria used to choose to remove or not a publication should be mentioned in the manuscript.~
The authors agree with the reviewer suggestion and the manuscript was corrected accordingly. It was added a sentence, in the toxicology section, referring the selection of publications for this review.
"A bibliographic search of the scientific literature was conducted on Google scholar using the search terms “ecotoxicology” and each of the selected compounds. All the publications that presented ecotoxicological studies on the selected compounds, referring the concentrations used were included.”
- Only one or two toxicity values are reported for a same species. For some compounds, it seems that these values are the lowest reported in the literature but this not the case for all the compounds. Explain the choice of the reported values.
The authors made a bibliographic search of the scientific literature on Google scholar using the search terms “ecotoxicology” and each of the selected compounds. All the publications that presented ecotoxicological studies on the selected compounds, referring the concentrations were included.
The authors included 120 publications regarding ecotoxicology data, which is a very reasonable number. If key publications are missing, the authors kindly ask for those references that we will gladly include in this review.
- L.125: What is a low concentration? Because these toxic concentrations are still some order of magnitude higher than environmental concentrations (with some exceptions).
The purpose of that statement is that we can observe toxic effects at low concentrations (ng/L or mg/L), we do not need high concentrations (mg/L or g/L) to promote ecotoxicological hazard, highlighting the toxicity of the selected compounds. Additionally, “considering all trophic levels, all therapeutic groups with the exception of anxiolytics, had at least one toxicity report for concentrations below 1 µg L-1”. If there is toxicity in the range of ng/L we are close to the concentrations found in the aquatic environment.
However, the manuscript was corrected to clarify the authors purpose.
- Avoid sentences stating the obvious (e.g. L.129-130, L.136-140, …). L. 139 “once again, as anticipated, LC50 data presented the highest concentrations”. By definition, LC50 values will always be higher than NOEC and LOEC values. Otherwise there will be a problem.
The authors agree with the reviewer suggestion and the manuscript was corrected accordingly.
- Some of the data for fishes correspond to data on cell lines or tissue. It is quite difficult to then extrapolate to the entire organism or even say that fishes are more sensitive to other organisms for which data were obtained on the entire organism.
The authors agree with the reviewer suggestion and the manuscript was corrected accordingly, by adding a sentence clarifying this limitation.
“The limitation of this analysis is that, regarding fish, there were also toxicity data obtained through cell line or tissue testing, which can be difficult to extrapolate to the entire organism.”
- L141: What do the authors mean by "do not produce visible results". Maybe there is no chronic toxic effect of these compounds at all. Please rephrase.
The authors agree with the reviewer suggestion and the manuscript was corrected accordingly by removing that expression.
That sentence purpose was to state that there are insufficient chronic data regarding the selected pharmaceuticals and there are some pharmaceuticals with no data at all.
- Figures 4-6: Risk quotients are classically calculated only using the most sensitive organisms, since by protecting the more sensitive obviously the other will be protected too. Thus, it is not necessary to calculate RQ for all the trophic levels. Otherwise, represent the data of toxicity as species sensitivity distribution and use hazard quotient 5%.
As in several other publications, the authors believe that the best way to present these results is to provide the environmental risk assessment for each trophic level. This way, it is possible to have an overview on the impact of pharmaceuticals contamination in different aquatic species, observing which trophic level is more sensitive for each therapeutic group/pharmaceutical. Providing a better understanding of ecotoxicological effects of the selected pharmaceuticals in the different aquatic compartments.
- Authors used data obtained on freshwater species to derive RQ for seawater whereas there are several publications dealing with marine organisms exposed to some of the selected compounds showing a strong sensitivity of marine organisms, much higher than freshwater organisms. And RQs below 1 have been calculated for antidepressants and antibiotics, among other compounds. The authors should modulate their text, or discuss this point.
The authors agree with the reviewer suggestion and the manuscript was corrected accordingly by adding this limitation in the ERA evaluation on in other water bodies.
“The results obtained in seawater can be biased, since fresh water organism were used to evaluate the risk and there are reports that marine organism can be more vulnerable, increasing the risk in this water matrix”
- I do not see the point to perform Environmental/Ecological risk assessment for drinking waters. I would rather understand human health risk assessment.
The authors understand the reviewer point of view, however, this review is on environmental risk assessment and therefore, human risk assessment was not undertaken.
The purpose of the authors was to compare the ERA for the three trophic levels in different aquatic environments, to understand the risk. Since most of the drinking waters are collected from groundwaters and surface waters, the inclusion of this type of water allowed to assess if there was any differences between these matrices.
Nevertheless, the manuscript was corrected clarifying the author’s idea.
“Since drinking water is usually obtained through surface and groundwaters, these results suggest that the sources used to produce drinking water are the ones with lower pharmaceutical contaminations or that water treatment plants are removing the selected pharmaceuticals.”
- Concentrations measured in the different aquatic environments are reported in a previous publication, but it will be easier if the authors can provide the concentrations used here to calculate RQs, in the present publication. The reader will also be able to better understand how RQs (mean – max RQs) were calculated (what were the data used?).
The authors agree with the reviewer suggestion and the manuscript was corrected accordingly by adding Table S2 to Supporting information with the requested data from Part I of this review.
- L463, 465: I do not agree. The author did not conduct a careful literature review or a complete overview of the state of the art, since too many publications are missing.
The authors made a bibliographic search of the scientific literature on Google scholar using the search terms “ecotoxicology” and each of the selected compounds. All the publications that presented ecotoxicological studies on the selected compounds, referring the concentrations were included.
The authors included 120 publications regarding ecotoxicology data, which is a very reasonable number. If key publications are missing, the authors kindly ask for those references that we will gladly include in this review.
However, the authors amended that sentence.
Minor comments:
- 21: Remove “and” at the end of the sentence.
The authors agree with the reviewer suggestion and the manuscript was corrected accordingly.
- 47: Replace “aquatic mesocosm” by “aquatic ecosystem”
The authors agree with the reviewer suggestion and the manuscript was corrected accordingly.
- Figures: Add the number of data used for each point (n= ?), and specify the meaning of the whiskers and the point (CI95% or min-max values? Means or medians?)
The authors agree with the reviewer suggestion and the manuscript was corrected accordingly.
Sup data:
Be consistent throughout the table:
- Species names are needed, since for a same phylum different species can have different sensitivities.
The authors agree with the reviewer suggestion and the manuscript was corrected accordingly. However, data from ECOSAR was not provided with species names, since it is a computerized predictive system that estimates aquatic toxicity based on Ecological Structure Activity Relationships.
- One line per species: all the data available for the same species should be written one after the other and not separated by the data of other species.
The authors agree with the reviewer suggestion and the manuscript was corrected accordingly.
- EC50 (72h) data for algae species are considered by the authors either as acute or chronic ecotoxicity data. First line of Table S1 for Alprazolam, the authors considered EC50 (96h) for green algae as an acute ecotoxicity data. Ecotoxicity tests at 72h are often considered as acute toxicity tests but considering the generation time of algae species, it can also be considered as chronic ones. So, the authors should choose acute / or chronic, and be consistent throughout the table.
The authors followed the classification of the original publications, however, the authors agree with the reviewer suggestion and the manuscript was corrected accordingly, both Table S1 and the manuscript text and figures. The OECD “Guidance document on the use of the harmonised system for the classification of chemicals which are hazardous for the aquatic environment” states that the toxicity test used for freshwater algae is the 201 and that the duration is “usually 3 to 4 days duration”, being considered a “short-term test”. Therefore, the authors considered acute toxicity data for algae when the test duration was until 4 days and chronic studies when were longer than 4 days.
Additionally, in the “Toxicity” chapter, criteria of inclusion in acute or chronic toxicity was provided.
- Some tests are missing (e.g. p.43 last line – réf 84 or p.47 – réf 96, …).
The authors agree with the reviewer suggestion and the manuscript was corrected accordingly.
Reviewer 4 Report
The review by Pereira and coworkers deals with pharmaceuticals present in different aquatic ecosystems and their toxicity. indeed, this is an interesting article and believed that increasing these molecules in aquatic environment s should carefully monitor to evaluate potential risks in biota. besides, it should be of a broad interest for environmental researchers. prior to acceptance, the following minor concerns should be addressed. All the chemical structures in this manuscript should be revised for the clarity. I would reccomend the authors to use software such as "ChemDraw". The size of the structure blocks should be identical and uniform. The length of the chemical bond should be suitable and I suggest the authors to use the style "ACS document 1996" or other approved one may be used. Line no 73 : Reference missing. Conclusion: Sediments may adsorb mor pollutants than that of surface water. I suggest the authors specify this in connection with their reviewed pharmaceuticals.
Author Response
The review by Pereira and coworkers deals with pharmaceuticals present in different aquatic ecosystems and their toxicity. indeed, this is an interesting article and believed that increasing these molecules in aquatic environment s should carefully monitor to evaluate potential risks in biota. besides, it should be of a broad interest for environmental researchers. prior to acceptance, the following minor concerns should be addressed.
All the chemical structures in this manuscript should be revised for the clarity. I would reccomend the authors to use software such as "ChemDraw". The size of the structure blocks should be identical and uniform. The length of the chemical bond should be suitable and I suggest the authors to use the style "ACS document 1996" or other approved one may be used.
The authors agree with the reviewer suggestion and the manuscript was corrected accordingly by using “ChemDraw”.
Line no 73 : Reference missing.
The authors agree with the reviewer suggestion and the manuscript was corrected accordingly.
Conclusion: Sediments may adsorb mor pollutants than that of surface water. I suggest the authors specify this in connection with their reviewed pharmaceuticals.
The authors believed that it was more suitable to address this issue in Part I of this review (Selected pharmaceuticals in different aquatic compartments: Part I - source, fate and occurrence) were the pharmaceuticals with higher log Dow (pH 7.4) and log Koc (pH 7.4) were identified. However, “some authors defend that the log Kow and log Koc approaches are excessive restrictive models of pharmaceuticals distribution in environment. In complex natural water and wastewater samples, partitioning due to hydrophobicity/lipophilicity is not the only physicochemical force of attraction operating between molecules. Electrostatic interactions, chemical bounding and non-specific forces between ionized molecules and dissolved organic matter are neglected through exclusive log Kow and Koc approaches.”
Additionally, “the physicochemical properties of pharmaceuticals show a high variability. For example, the log Dow ranges from -2.23 to 4.6, the log Koc varies between 0 and 3.88 and even solubility goes from 0.1 to 101 200 (mg L-1). These variations are not only observed between different therapeutic groups, but also within each group, since, as previously referred, this pharmaceuticals grouping does not correspond to similar chemical structures and there are more than one class per group. This can be seen especially for antibiotics, lipid regulators and anti-inflammatories, where greater fluctuations in these parameters are reported.
In summary, although pharmaceuticals present different physicochemical properties, some are expected to be more lipophilic and others to sorb to soils and sediments, they all have relatively high water solubility, having the potential to contaminate the aquatic environment”
Round 2
Reviewer 3 Report
The manuscript has been improved. I have no more comments, as the authors have carefully replied to my previous comments and made the changes on their manuscript accordingly.